# Marginal Adaptation and Micropermeability of Class II Cavities Restored with Three Different Types of Resin Composites—A Comparative Ten-Month In Vitro Study

**DOI:** 10.3390/polym13101660

**Published:** 2021-05-20

**Authors:** Sevda Mihailova Yantcheva

**Affiliations:** Medical University-Sofia, Faculty of Dental Medicine, Department of Conservative Dentistry, 1, G. Sofiiski Bvd., 1431 Sofia, Bulgaria; sevda.yancheva@gmail.com

**Keywords:** marginal adaptation, microleakage, nanocomposite, silorane, SonicFill

## Abstract

The development of composite materials is subject to the desire to overcome polymerization shrinkage and generated polymerization stress. An indicator characterizing the properties of restorative materials, with specific importance for preventing secondary caries, is the integrity and durability of marginal sealing. It is a reflection of the effects of polymerization shrinkage and generated stress. The present study aimed to evaluate and correlate marginal integrity and micropermeability in second-class cavities restored with three different types of composites, representing different strategies to reduce polymerization shrinkage and stress: nanocomposite, silorane, and bulk-fill composite after a ten-month ageing period. Thirty standardized class ΙΙ cavities were prepared on extracted human molars. Gingival margins were 1 mm apical to the cementoenamel junction. Cavities were randomly divided into three groups, based on the composites used: FiltekUltimate-nanocomposite; Filtek Silorane LS-silorane; SonicFill-bulk-fill composite. All specimens were subjected to thermal cycles after that, dipped in saline for 10-mounds. After ageing, samples were immersed in a 2% methylene blue. Thus prepared, they were covered directly with gold and analyzed on SEM for assessment of marginal seal. When the SEM analysis was completed, the teeth were included into epoxy blocks and cut longitudinally on three slices for each cavity. An assessment of microleakage on stereomicroscope followed. Results were statistically analyzed. For marginal seal evaluation: F.Ultimate and F.Silorane differ statistically with more excellent results than SonicFill for marginal adaptation to the gingival margin, located entirely in the dentin. For microleakage evaluation: F.Ultimate and F.Silorane differ statistically with less microleakage than SonicFill. Based on the results obtained: a strong correlation is found between excellent results for marginal adaptation to the marginal gingival ridge and micropermeability at the direction to the axial wall. We observe a more significant influence of time at the gingival margin of the cavities. There is a significant increase in the presence of marginal fissures (*p* = 0.001). A significant impact of time (*p* < 0.000) and of the material (*p* < 0.000) was found in the analysis of the microleakage.

## 1. Introduction

The dental practice uses composite materials for more than fifty years. During this period, they have undergone significant development. Their progress gradually imposes them as durable restorative materials [1,2]. Moreover, they allow one to meet the growing demands of patients, both in terms of functional and anatomical restoration of the dentition, and to achieve excellent aesthetics [3].

Dental resin composite materials consist of four main components: (1) organic polymer matrix (dispersed medium); (2) inorganic filler (dispersed phase) fillers and tins; (3) coupling phase that adheres the matrix to the filler particles (silanes); (4) activators and inhibitors of the polymerization process [4].

They are inhomogeneous in terms of structure, and their properties generally surpass the mechanical sum of the properties of the individual components [5].

One of the main problems in dental composites is polymerization shrinkage and generated stress caused by the shrinkage [6,7]. Therefore, the development of composite materials is subject to the desire to overcome these drawbacks.

The main reason for shrinkage is the polymer matrix of dental composites [4]. The resin matrix constitutes about 20–40 wt % of a resin composite material. It is composed of dimethacrylate monomeric compounds, including mainly Bis-GMA (bisphenol-A glycidyl dymethacrylate), UDMA (urethane dimethacrylate), TEGDMA (triethylene glycol dymethacrylate), and Bis-EMA (ethoxylated bisphenol-A glycidyl dymethacrylate) [8]. Variable combinations and proportions of these monomers are included in current resin composite materials resulting in different copolymer systems. These monomers have a linear structure. During free-radical polymerization, the molecules move towards each other, bonding with covalent bonds. The intermolecular distances are smaller in the obtained polymer network than in non-polymerized molecules—volumetric shrinkage occurs [5,8]. When the material is limited and connected to the rigid walls of the cavity, the shrinkage forces increase even more and create tension in the composite–tooth connection [9,10]. The developed polymerization stress can lead to rupture of the adhesive bonds with the hard tooth tissues. Consequently, the appearance of cracks, allowing microleakage, hypersensitivity, and lately secondary caries, damage to the dental pulp can occur. If the adhesive bond is strong enough and withstands stress, cohesive fractures in the composite, deformations in the tooth walls, and formation or growth of cracks in the enamel may occur [11,12,13].

There are two main strategies to reduce polymerization shrinkage [14]:By reducing the reactive groups per unit volume of the polymer matrix.This goal can be achieved by:
(a)Increasing the relative share of filler particles in the inorganic phase.(b)Increasing the molecular weight of the reactive groups in the organic matrix.By using a different type of organic matrix.

Historically, efforts to reduce shrinkage have focused primarily on increasing the proportion of the inorganic phase through changes in particle size, shape, and distribution. Development of the inorganic phase made the strength and toughness of dental composites comparable to dental amalgam and porcelain [15], expanding the indications of composites as a material durable for restorations in the distal, stress-bearing zone of the dentition. One of the latest achievements in inorganic component progress is the penetration of nanotechnology in developing and improving dental composites. Nanotechnologies are involved in developing inorganic fillers [16,17], enabling the organic phase volume to be reduced due to the possibility of even greater particle saturation. Nanofilled composites contain silica/zirconia individual nanoparticles (5–20 nm) and fused/agglomerated nanoclusters (average size 0.6–10 μm). Nanohybrid composites contain silica/zirconia nanoparticles and larger 0.6–1 μm glass/silica/zirconia particles. Some of them can contain prepolymerized resin fillers. Nanoparticles are responsible for excellent optical properties and polishability [3,16,17]. Hardness, strength, and wear resistance are similar to microhybrid composites [18,19]. An average reduction of the polymerization shrinkage to about 1.5–2.5% is established [20].

The change in the polymer matrix of dental composites has been worked on more successfully in the last decade. Finally, it comes to the creation and implementation of a dental composite with a completely different matrix. This material is silorane. Its molecule is a siloxane centre connected by four oxirane rings, which open during polymerization to bind to other monomers [4,8,14]. The cationic ring-opening reaction reduces the shrinkage below 1%. Mechanical properties of silorane are similar to methacrylate-based composites [14]. However, due to the reduced volumetric shrinkage, the silorane composite leads to less tooth deformation [10,21,22,23] and micropermeability [24].

Apart from the polymerization shrinkage associated with dental composites, another problem is the sensitive and long-lasting clinical protocol for building the restorations. In addition, the performance of class I and II filings usually require layer application of a material. Therefore, the layering technique is necessary for two main reasons:Light-cured composites have a limited polymerization depth—2 mm.The polymerization of dental composites is a complex process, depending on their composition and heterogeneous structure. Manufacturers of dental resins give recommendations about the depth of cure as they relate to light activation. The most typical indication is the use of specific light intensity and exposure time, which can cure 2 mm of material. Depth of cure of 2 mm provides maximum conversion rate, hardness, and the composite material’s stability [25,26].Clinicians try to control the polymerization shrinkage stress of the material by incremental application of the composite. The concept of layer-by-layer application of composites is based on applying a small volume of material with minimal contact with the opposite walls of the cavity (C-factor) during polymerization. It has been found that the small volume of the composite causes less shrinkage [27]. Theoretically, each layer is compensated by the next, and the overall volumetric shrinkage is less destructive because the free surfaces allow stress release by providing flow [9,27].

Therefore, in the case of deep or extensive preparations, more than one layer of material must be applied, which is time-consuming and involves certain risks such as the formation of air bubbles and contamination between the layers [8,28,29].

In response to these difficulties, a new generation of composites was introduced, known as bulk-fill composites. Manufacturers use this term to denote composites that can be used in layers with a thickness of 4 or 5 mm through a single-layer (monoblock) technique [29].

The manufacturers claim that the bulk-fill composites have reduced polymerization shrinkage than the flowable composites and the conventional composites for layer application [28]. Bulk-fill materials have a dimethacrylate matrix and include in their composition specific modifiers of the polymerization process. They are reports of decreased polymerization shrinkage stress, reduced cusp displacement connected with high C-factor cavities and good bond strength related to bulk-fill composites [30,31,32].

An indicator characterizing the properties of restorative materials, with particular importance for preventing secondary caries, is the integrity and durability of marginal sealing. Thermocycling and ageing with SEM marginal analysis and microleakage assessment are still closer to the clinical situation, allowing us to compare these important for the clinic characteristics of different restorative materials [15].

The present study aimed to evaluate and correlate marginal integrity and micropermeability in second-class cavities restored with three different types of composites: nanocomposite, silorane, and bulk-fill composite after a ten-month ageing period.

## 2. Materials and Methods

A schematic description of the experimental design can be seen in Figure 1.

### 2.1. Specimen Preparation

Fifteen extracted intact human third molars were used in this study. The teeth were extracted for orthodontic purpose and collected with patients informed content. The teeth were cleaned with a hand curette of soft tissue residues, washed with running water and stored for no more than two months in distilled water with added thymol crystals. Before the cavity preparation, the teeth were rewashed with running water and a toothbrush. Thirty standardized class II cavities (MO/DO) were prepared. The parameters of the cavities were approximately the following: vestibular-lingual size = 3 mm; medial-distal size = 2 mm; axial size = 5 mm. The gingival wall was located 1 mm below the cementoenamel junction. The edges of the cavities were not bevelled. The cavities were prepared with turbine cylindrical burs with a rounded tip (Z880-140-FG012M-NTI; NTI-Kahla, Germany).

The vestibular and lingual walls of the cavities were approximately parallel to each other. After cavity preparation, the teeth were divided randomly with 10 (*n* = 10) cavities per studied material.

Before the cavities were restored, the teeth were included in silicone models with adjacent phantom molars. Study simulated conditions close to the clinical ones in the production of composite restorations.

Filtek Ultimate (3M ESPE, St. Paul, MN, USA) and SonicFill (Kerr, Orange, CA, USA) (dimethacrylate based resin composites) were applied after using the same three-step adhesive system Optibond FL (Kerr, Orange, CA, USA).

Filtek Silorane (3M ESPE, St. Paul, MN, USA) was applied after using Silorane System Adhesive—Primer and Bond (3M ESPE, St. Paul, MN, USA), the only adhesive system used with the silorane material.

The adhesive systems were applied according to the manufacturer’s instructions and after placement of a contoured metal matrix (MetaFix-Kerr, Kerr, Orange, CA, USA) and a wooden interdental wedge.

Filtek Ultimate and Filtek Silorane were inserted in 2 mm layers: the first horizontal and the next two oblique. Each layer was polymerized for 40 s.

SonicFill was applied at ones in bulk in about a 5 mm portion and polymerized for 40 s. The material was applied with a specially created SONICfill handpiece (KaVo/Kerr, Germany) and uses sound energy to lowers composite viscosity.

A diode photopolymer lamp (Elipar Freelight II; 3M-ESPE, St. Paul, MN, USA) was used. Soft start mode of photoactivation was applied.

After the restoration of the cavities, finishing and polishing were performed with abrasive discs (Sof-lex/3M-ESPE, St. Paul, MN, USA), polishing paste, and rubbers.

The main composition of resin-composite materials included in the present study is described in Table 1.

### 2.2. Termocucling, Ageing, and Preparation for Microscopic Investigation

After completing the restorations, the teeth were removed from the models, cleaned, and washed under running water and a toothbrush. The teeth were placed in saline for 24 h, then thermocycled for 1000 cycles (5–55 °C for 1-min immersion in each bath with a 30 s interval between immersions). Thermocycling apparatus T H E 1100/1200 was used.

After the thermal loading, the samples were placed in saline for ten months and stored at room temperature. Every two weeks, the solution was renewed. After ten months, the samples were removed from the solution, washed, dried, and apexes sealed with adhesive wax. The restorations were insulated 1 mm from their border with nail polish and stained for 12 h in 2% methylene blue. After removing the dye, the teeth are washed under running water for 20 min and cleaned with polishing toothbrushes.

#### 2.2.1. SEM Investigation—Marginal Integrity

The samples thus prepared were coated with gold by low vacuum cathodic sputtering (JEOL JFC-1200) and observed on a scanning electron microscope to directly evaluate the marginal seal of the restorations (SEM-JEOL JSM-5510).

The peripheral connection was examined sequentially for each marginal edge. Magnifications of 25× and 250× were used.

The measurements have proceeded with KLONK Image Measurement 14.1.1.4, Copyright 2013, Klonk Sm Ba.

Estimates are presented as a percentage of the total length of the relevant edge.

The following ratings were given [33]:Excellent margin;Over-filled margin; Under-filled margin;Marginal fissure.

The rating scale is illustrated in Figure 2.

#### 2.2.2. Stereomicroscopic Investigation—Micropermeability

When the SEM analysis was completed, the teeth were included into epoxy blocks and cut longitudinally on a Leica SP 1600 microtome at 1 mm slice thickness on three slices for each cavity. Usage of 3 slices allowed analyzing four surfaces of the restoration, giving information of the level of penetration of dye in the direction of the axial wall of the cavity. An assessment of microleakage followed. It was performed on a stereomicroscope (Leica MZ6) at 40× magnification.

The penetration level of the dye was assessed on the following scale [34]:0.No due penetration;1.Due penetration up to 1/3 of the gingival wall;2.Due penetration up to 1/2 of the gingival wall;3.Due penetration over 1/2 of the gingival wall, but the axial wall is not affected;4.Due penetration reaches and covers the axial wall of the cavity.

### 2.3. Statistical Analysis

The results obtained from both studies were statistically processed. The level of statistical significance is considered to be *p* ≥ 0.005. The IBM SPSS Statistics statistical package was used for statistical data processing. ANOVA (analysis of variance), Chi-square analysis, and post-hoc LSD analysis were performed. Based on the results of studies, a correlation was sought between the marginal seal and the micropermeability in the direction of the axial wall—a Spearman correlation coefficient was used.

The impact of ageing on marginal adaptation and micropermeability was established through a comparative analysis of early data recorded by our team. The same experimental setup was used [35].

## 3. Results

### 3.1. Marginal Integrity—Results

Table 2, Table 3 and Table 4 present the results of the study of marginal adaptation. Table 2 shows the summary data on estimates for the edge of the gingival wall, located entirely in the dentin (1 mm below the cementoenamel junction CEJ). In terms of excellent marginal adaptation to the edge of the gingival wall, the best indicators have the restorations made by F.Ultimate (41.61%), followed by F.Silorane (41.24%). In the filings made by S.Fill, there were fewer specimens with excellent marginal adaptation to the gingival margin. There are the most registered results for the over-filled margin on the gingival edge of the restoration (S.Fill—48.84%). For restorations made by F.Ultimate and F.Silorane, there is less evidence for over-filled gingival margins. Less evidence for marginal fissure presence is reported for F.Silorane restorations.

In Figure 3, the mean results of the marginal adaptation to the marginal gingival edge are presented graphically.

A comparison was made between the three materials on marginal adaptation to each of the estimates’ proximal edges (analysis of variance). The average levels of the indicators were used. The study found a statistically significant difference between the materials. In addition, a difference exists in terms of marginal adaptation to the edge of the gingival base (Table 2). The differences are in terms of the following scores: excellent marginal adaptation (*p* = 0.005). The post-hoc analysis showed that S.Fill differed significantly from the group of F.Ultimate and F.Silorane with more unfavorable results.

There is a statistically significant difference in the estimates for marginal adaptation of the materials to the vestibular and lingual proximal edges (Table 3 and Table 4): excellent marginal adaptation to the vestibular marginal edge (*p* = 0.012 < 0.05); excellent marginal adaptation to the lingual marginal edge (*p* = 0.025 < 0.05); the presence of marginal fissures relative to the lingual marginal edge (*p* = 0.024 < 0.05). The post-hoc analysis cannot determine with statistical reliability the differences between the materials, both for the excellent results and the results for the presence of marginal fissures, concerning the two enamel edge.

In Figure 4, the results of the marginal adaptation to the vestibular/lingual proximal borders are represented graphically. The results for both margins were averaged.

### 3.2. Micropermeability—Results

The results of the micropermeability study are presented in Table 5.

Leakage involving the axial wall of the cavities was observed. Filtek Silorane is an exception in this respect. There is no penetration reached the axial wall of Filtek Silorane specimens.

According to the summarized results for each of the materials, the post-hoc analysis showed that Filtek Silorane differed significantly from SonicFill (*p* < 0.001). The difference between Filtek Silorane and Filtek Ultimate was not significant (*p* = 0.280).

Filtek Ultimate differed significantly from SonicFill (*p* = 0.05).

Filtek Ultimate was on par with Filtek Silorane with better results.

### 3.3. Correlation between Marginal Integrity and Microleakage

The five values of micropermeability for each sample of each material were averaged. Then, the correlation coefficients (according to Spearman) between the micropermeability and the excellent estimates for marginal adaptation to the proximal edges of class II cavities were calculated. There was a significant correlation (*p* = 0.008) between the excellent scores for marginal adaptation to the edge of the gingival wall and micropermeability: correlational coefficient = −0.338.

There are no significant correlations between the excellent scores for vestibular and lingual proximal margins and micropermeability to the axial wall of the cavities.

Conversely, following the correlation coefficients between marginal fissure estimates at the marginal proximal edges of class II cavities and the degree of micropermeability, no significant correlations were found anywhere.

### 3.4. Comparison between Early and Late Results (Ageing and Material Influence)

This experimental design was employed completely repeating experimental conditions and the methodology of our previous study. The prior investigation recorded the results of the experiment immediately after thermal cycling. The specimens were not aged. The results of the last survey have already been published [35]. The same experimental design was employed because we wanted to determine the influence of ageing on marginal adaptation and micropermeability.

Two-factor analysis of variance with independent variables “material” and “time of results” (late and early) was performed separately for each edge and each evaluation.

We observe a more significant influence of time at the gingival margin of the cavities. The experimental setup of the study was located entirely in the dentin. In the late results, we registered a reduction of excellent assessments for the quality of the marginal adaptation compared to the early ones (*p* = 0.084). In addition, there was a significant increase in marginal fissures in the late compared to the earlier results (*p* = 0.001). A comparison of results is presented in Table 6.

The cavity edges located in the enamel—the vestibular and lingual proximal edges—also increased the gaps in the late compared to the early results (*p* = 0.084/*p* = 0.089). Still, we did not report statistically significant changes in the excellent assessments of the marginal contour (*p* = 0.102).

Despite less excellent results in the gingival margin region registered for SonicFill in early results, statistical analysis did not find significant differences between materials regarding these values.

Data from early and late results were compared to establish the influence of ageing on micropermeability.

A variance analysis was performed comparing the early and late results of the studies of the three materials. A significant influence of time F (1.468) = 23.732, *p* < 0.000, and a significant influence of the material F (5.468) = 7.243, *p* < 0.000 was found. The comparison between results is represented in Figure 5.

In the early results, we should note, F.Silorane differed significantly from other materials with values (0) lack of due penetration. None of the materials registered leakage involving the axial wall of the cavity. A significant difference in favour of F.Silorane with better results was recorded compared to SonicFill. F.Ultimate did not differ statistically from F.Silorane or SonicFill.

## 4. Discussion

We expect dental materials to be long lasting in the oral cavity. Composite restorations must withstand complex conditions that vary, including depending on the patient—dietary factors, chugging habits, saliva and enzymatic activity, thermal fluctuation, and oral hygiene. To predict and mimic the influence of oral conditions, the restorations of different materials in the present study were prone to thermal changes and ageing for ten months in a wet environment. Thermal stress attacks the marginal integrity of restorations due to the different coefficient of thermal expansion and contraction of hard dental tissues and composite materials. It leads to stimulation of percolation, fatigue in the adhesive bond area, to the appearance or increase of marginal permeability [36,37,38].

Under the impact of liquid medium, hydrolytic expansion, dissolution, mechanical erosion, degradation, and softening occur [39,40]. These processes act on the quality of filings, affect the weaknesses of marginal adaptation and resilience over time [41]. The present experiment results found clear support for the influence of oral simulating conditions on the composite restoration qualities. The percentage of excellent values of marginal adaptation decreases, and the presence of marginal fissures increase over time. The marginal gingival edge of the restorations was most significantly affected. Here the expansion of the gaps was statistically significant, and the tendency to decrease the zones of excellent adaptation of the materials show a clear direction to reduce. Figure 6 makes an illustrative comparison between early and late results for the individual materials. It can be seen that the restoration edge stands out more, most likely due to hydrolytic expansion. In the late results, more defects in marginal adaptation are easily seen.

In clinical conditions, one of the main reasons for the replacement of class II restorations is secondary caries, which occur in the area of the marginal proximal ridge [25,26]. The most affected is the marginal gingival ridge due to the disturbance of the tooth-restoration connection and the increased levels of cariogenic microorganisms in this area of the filing [26].

In the proximal gingival ridge of class II cavities, the enamel is often incomplete or absent. To avoid the questionable significance of cervical enamel on the integrity of the marginal seal, the cavities included in the present study were prepared with a gingival wall 1 mm apically from the cementoenamel junction. These are complex cases in clinical practice—the gingival wall with its adjacent outer edge is located entirely in the dentin. The adhesive bond to the dentin is weaker than that to the enamel due to the higher organic content of the dentin tissue, the orientation of the dentinal tubule, difficult elimination of the smear layer and moisture [42,43]. The polymerization shrinkage in this area toward the light source and weaker bond with the dentine can more easily lead to the formation of gaps between the cavity margin and the composite material (occurrence of an open peripheral connection) with subsequent secondary caries. This fact is evidenced by the current results, which show a significant effect of ageing on the marginal adaptation of composites to the marginal gingival edge, complemented by an increase in microleakage to the axial wall of the cavity.

Direct sputtering with gold under the low vacuum for evaluating the marginal adaptation allows this study to be combined with the micropermeability analysis by the dye penetration method. Two indicators, essential for assessing the integrity and tightness of restorations, can be examined on the same samples, and a correlation between the two studies can be sought. We found a significant correlation between the excellent scores for the marginal gingival seal and the micropermeability (correlational coefficient −0.338). The dependence increases over time. Materials with more excellent gingival marginal seal scores: F. Silorane (41.24%) and F. Ultimate (41.61%) showed lower levels of micropermeability.

A similar conclusion was reached by Heintze et al. [44]. They, too, establish a relationship between data on the restorations’ marginal seal and the level of micropermeability (class II cavities with a gingival base below CEJ).

Ernst et al. [34] find a good correlation between the SEM data and the microleakage in the study concerning the marginal seal of the restorations (class V). However, they found a correlation at the boundaries in the enamel but not at the dentin. Contrary to Ernst et al., we did not find significant correlations between the excellent scores for the enamel margins (vestibular and lingual) and micropenetration. Thus, the results involving Class V cavities may not reflect the leakage pattern of Class II cavities, which are larger and more complex.

Following the correlation coefficients between the estimates for marginal fissures (open peripheral connection) and the degree of micropermeability, we did not find significant correlations anywhere. A similar result was obtained by Ishikirima et al. [45]. They did not find a relationship between the marginal gaps and the degree of microleakage. Some essential publications concerning secondary caries demonstrated that the presence of marginal gaps in vivo does not indeed have to be accompanied by secondary caries [46].

Analyzing the results from the present study, we could conclude that for the degree of micropermeability in class II cavities with a gingival wall below the CEJ, a key role included the excellent marginal adaptation of the composite to the marginal gingival ridge. Overall, these findings are following conclusions reported by a six-year, parallel in vitro/in vivo study [41]. Thus, on the one hand, researchers established a correlation between the results of laboratory and clinical studies. On the other, it demonstrated that the continuous contour of class II restorations is crucial for the clinical durability of the filings.

The question remains to what extent the excellent and continuous marginal contour must be disrupted to lead to clinical complications.

Materials with better marginal adaptation to the marginal gingival ridge and lower micropermeability are nanocomposite and silorane composite. This data shows that the development of resin composite compositions by changing the inorganic phase through nanotechnology and changing the matrix of materials with silorane leads to comparable results over time.

Filtek Ultimate (3M ESPE) is representative of nanocomposite materials. It contains particles of 4–20 nm and nanoclusters 0.6–20 µm. The reduction of the polymerization shrinkage for Filtek nanocomposites is up to 1.67%, regarding scientific investigations [47]. Present study results for the nanomeric composite are comparable and in some aspects superior (better marginal adaptation to vestibular and lingual enamel margins) to the impacts of the low shrinkage silorane matrix composite. We could find a possible explanation for this finding in the more plastic texture of F.Ultimate compared to the F.Silorane, which allows better marginal adaptation to the cavity elements, providing improved wetting ability. The denser state of silorane composite is due to its matrix properties, not because of high filler content [21].

Another benefit of nanotechnology favorably influencing marginal integrity is lower polymerization stress because of round-shaped filler particles [18,48]. Furthermore, fillers can distribute the load stress and inhibit crack propagation due to pinning effects [18]. Shibasaki et al. [47] investigated mechanical properties and curing behavior of different types of modern resin composites. For F. Suprime, a precursor of F.Ultimate, they stated the balance of filler size, distribution of fillers, and the interaction between fillers and matrix through silane coupling agent contribute effectively to the dispersion of load stress. This conclusion complements the explanation of the reported good results registered by us for the nanocomposite in marginal adaptation and micropermeability.

A study by Santos et al. [49] analyzed the properties of marginal adaptation of Filtek Ultimate and Filtek P60 in cavities with different C-factor. Results showed that regardless of the similar composition of the matrix, the differences in the filling phase and the modulus of elasticity (11.7 GPa Filtek P60/10.5 GPa Filtek Ultimate) lead to different behaviours concerning polymerization stress and gap formation. The nanometer composite, which has a lower modulus of elasticity, generates less stress and minor marginal defects.

However, in the present study, including the ageing of 10 months in saline, the number of gaps increased significantly, especially in the gingival margin for F.Ultimate (3.49% before ageing/35.37% after ageing). This result is in correlation with studies showing that nanocomposites deteriorate under in vitro conditions simulating ageing. Furthermore, deterioration of nanocomposite properties after storage in water has been reported [16,50]. It has been suggested that this may be due to the increased space between the matrix and the ultra-small particles, which leads to increased water sorption and degradation of the matrix/particle bond [51].

De Moraes et al. [52] reported significantly lower sorption of a nanocomposite than a hybrid and other nanohybrid composites tested. All tested composites displayed comparable solubility values. The authors suggest that water sorption and solubility results are probably related to the organic matrix chemical nature rather than the filler content of the material. Based on the present study results, it can be hypothesized that solubility characteristics of materials influence more marginal adaptation and micropermeability of the restorations than water sorption. The current results of the nanomaterial over time speak of its stability. Even in the case of late micropermeability results, it forms together with silorane the group of materials with lower levels of micropermeability.

The dye micropermeability test shows the sealing ability of the adhesive and composite complex.

A research team compared Filtek Silorane and Filtek Ultimate in class II cavities with a gingival base located in the dentin and found no significant difference in the two materials’ results in either enamel or dentin, despite the better performance of Filtek Silorane [53]. They conclude that the layering technique and correct selection of adhesive systems are crucial for the quality of restorations. In this study, the three-stage OptiBond FL (Kerr) was used as an adhesive with F. Ultimate. According to scientific data, three-step adhesives can build a good enough connection to enamel and dentin [53,54]. In addition, durable bonding ensures maximum adhesive safety of the composite fillings.

Filtek Silorane (3M-ESPE) is another material that shows acceptable, marginal adaptation to proximal margins in the present investigation. This fact can be attributed to the reduced polymerization shrinkage of silorane (lower than 1% determined by Archimedes method) and the different polymerization reaction of the material. For the polymerization of F.Silorane, it is necessary to accumulate a critical mass of initiators of the polymerization process. This demand leads to a slower and different course of the polymerization reaction. Additionally, the polymerization profile of the material differs from that of methacrylate composites [21,55,56]. Most likely, these features lead to a slower and more even distribution of the polymerization stress, reflecting on the preservation of the marginal integrity of the restoration, despite the higher modulus of elasticity of silorane composite (6.8 GPa) [21].

F.Silorane has a lower coefficient of linear thermal expansion [47]. This property of the material protects inducing gaps in the vicinity of the tooth/resin interface, resulting in less penetration of liquids between the restoration and cavity walls. Further, lower thermal expansion maces F.Silorane more stable during thermal loading. This behaviour is evident by the results registered for the material specimens immediately after thermocycling. F.Silorane was the only material with 17 scores (0) no due penetration (*n* = 40), and according to this indicator, it differs significantly from all materials tested [35].

F. Silorane is the material in which there is no registered penetration of the dye, engaging the axial wall of the cavity after ten months of ageing in a liquid environment. Probably, the proven high hydrolytic stability due to the hydrophobicity of silorane also contributes to these results [47,55,57].

The fact that silorane does not stand out statistically from the dimethacrylate nanocomposite after ageing may be due to shortcomings in the adhesive system of the material. The adhesive system of silorane is a two-step self-etch adhesive. It still possesses features of conventional methacrylate adhesives in terms of the bonding mechanism to tooth tissues. Still, the changes were needed to make it comparable with the high hydrophobic silorane matrix [58]. Research found that perhaps the existing difference in the chemistry of silorane and its adhesive system allows defects between the bond and the composite [59]. Spectroscopic examination established an intermediate zone of 1 μm between the primer and the silorane bond, explaining the higher percentage of adhesive fractures for the silorane groups [39]. Mention is made of the need for more prolonged monitoring of the durability of the hybrid layer built between the silorane composite and the dental tissues [60]. From the present study, the absence of a separate step of enamel etching in the adhesive system of silorane most likely leads to less excellent results concerning the vestibular and lingual edges of the examined cavities. However, this fact does not affect the microleakage, as found. Literature data show that the strength of the adhesive bond of silorane with hard dental tissues can be increased by using a separate step of etching the enamel, reaching values of 27.7 MPa [59]. Higher adhesive strength is due to the possibility of deeper inter- and intraprismatic penetration of the material [60,61,62].

Other researchers have also found an increase in marginal gaps in restorations made of silorane composite after a 6-month stay in a liquid medium (water). At the same time, they did not show a reduction in bond strength between the material and the preparation walls [39].

One of the innovative materials included in this study is the bulk-fill composite SonicFill (Kerr). When applying sound energy (5–6 kHz) through a tip, the modifiers of the polymerization process included in the matrix lead to a sharp decrease in viscosity (up to 87%). As a result, the flowability of the composite is increased, which allows fast filling and precise adaptation to the cavity elements [63]. Furthermore, the material regains its denser state upon cessation of sound energy, suitable for modelling and contouring [63]. SonicFill can be applied all at once in a 5 mm layer without increasing the polymerization stress [64,65,66]. At the same time, the new material is heavily filled—84% by weight. Due to these modifications, a low volume shrinkage was achieved—1.6% [66].

According to the unfavourable results of SonicFill from the present study, it is statistically distinguished from the nanocomposite and silorane. The more unsatisfactory results of marginal adaptation to the edge of the gingival base are associated with more unfavourable results in terms of micropermeability. On the other hand, the bulk-fill composite shows better adaptation to the enamel proximal edges than the gingival wall’s edge. This finding is confirmed by other scientific publications [30,67].

It has already been mentioned above that the material is highly filled (≈84% by weight). The high content of particles has led to the formation of a high modulus of elasticity—12.9 GPa [9,10]. Therefore, SonicFill is the composite with the highest modulus of elasticity included in the present study. However, the polymerization of all the material in the proximal cavity at once leads probably to more significant polymerization stress, associated with more unsatisfactory marginal adaptation and micropermeability.

Research on polymerization behaviour and mechanical properties of high viscosity bulk-fill and low shrinkage dental composites established that linear thermal expansion of bulk-fill materials is higher than the other categories [47]. In contrast, low-shrinkage resin composites showed significantly lower thermal expansion than the other resin composites. A large difference in thermal expansion induces gaps in the resin/tooth interface, resulting in the penetration of oral fluids between the restoration and the tooth surface. In the study of sorption and solubility, SonicFill ranked between F.Silorane and F.Ultimate in water sorption. In terms of solubility, SonicFill showed a higher solubility compared to F.Silorane and F.Ultimate [50]. This indicator probably also contributes to the deterioration of the properties of SonicFill restorations, subjected to 10 months of ageing in a liquid medium. It was hypothesized that it is possible large glass filler particles in SonicFill create routes for crack propagation between fillers and resin matrix [47].

Literature sources reported that SonicFill shows better adaptation to dentin in class II cavities with a base positioned apically by the ECJ, compared to other bulk-fill composites. However, there is no registered improvement in the properties of the peripheral connection compared to composites for layer application [66]. Our results confirm these findings.

The micropermeability is strongly influenced by the C-factor and the volume of the composite. This relationship is evident from research using microcomputed X-ray tomography [68]. It was found that the larger volume of the material and the larger C-factor lead to an increase in micropermeability. This conclusion confirms our results, recorded for SonicFill, which is applied at once in the cavity.

As Ferracane stated, “Placing Dental Composites is A Stressful Experience” [69]. Studies showed that the clinician’s influence on the quality of restorations is considerable and even decisive [1,70]. The development of materials with improved properties should reduce the operator’s influence and provide more predictable reliability of composite restorations. The elaboration of nanomaterials, silorane composite, and bulk files is a step in this direction. The present study showed that the materials requiring layering application, namely the nanocomposite and silorane, allow better marginal adaptation, limiting the restorations’ micropermeability. Bulk-fill composite using the monoblock technique and reducing more clinicians influence showed less satisfactory performance in those indicators necessary for the prevention of secondary caries.

The structure of dental composites is complex and requires the achievement of an appropriate balance between the components forming the final product, which will interact with the hard dental tissues and be exposed to the aggressive impact of the oral environment.

It is essential to understand the behavior of composite materials in laboratory conditions, which try to mimic realistic clinical situation. These investigations can help predict the restorative materials’ durability and select the optimal combination of cavity size, C-factor, location of margins, and layer thickness for different types of resin composites. In this connection, the obtained during the presented study results strongly suggest that additional well-designed investigations of various proximal cavity depths and positions of the gingival margin have to be provided with included mechanical loading. Furthermore, the question remains to what extent the excellent and continuous marginal contour must be disrupted to lead to clinical complications.

Long term investigation of sorption and solubility of new types of resin composites is needed, along with degradation pattern assessment.

## 5. Conclusions

According to the present study and the limitations arising from in vitro investigations, we can make the following conclusions:F.Ultimate and F.Silorane differ statistically with more excellent results than SonicFill, for marginal adaptation to the gingival margin, located entirely in the dentin in class II cavities, after ten months of ageing in saline.F.Ultimate and F.Silorane differ statistically with less microleakage than SonicFill in class II cavities with gingival margin located entirely in the dentin.There is a strong correlation between excellent results for marginal adaptation to the marginal gingival ridge and micropermeability at the direction to the axial wall in class II cavities, with a gingival wall 1 mm below CEJ.Ageing affects most significantly the proximal gingival margin, complemented by an increase in microleakage to the axial wall of the cavity.Present data shows that the development of resin composite compositions by changing the inorganic phase through nanotechnology and changing the matrix of materials with silorane leads to comparable results over time.

## Figures and Tables

**Figure 1 polymers-13-01660-f001:**
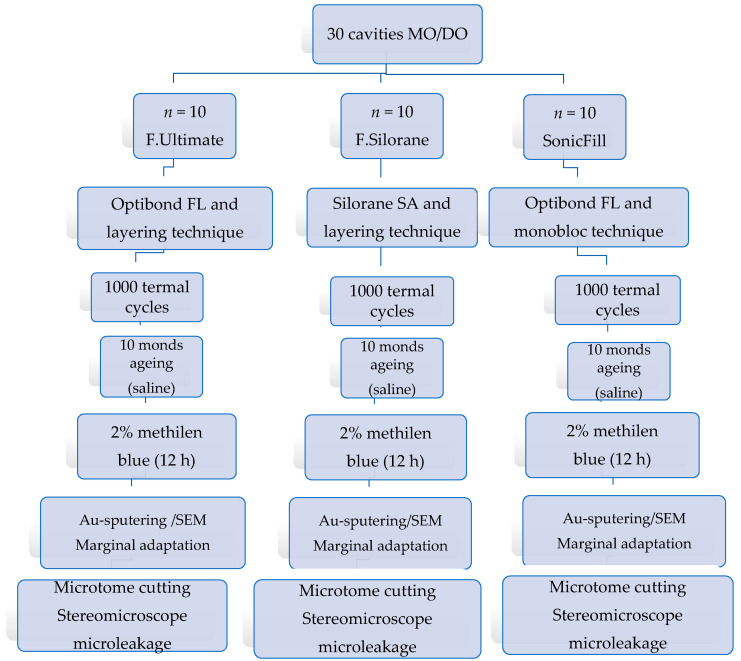
Schematic representation of the overall experimental design used in the study.

**Figure 2 polymers-13-01660-f002:**
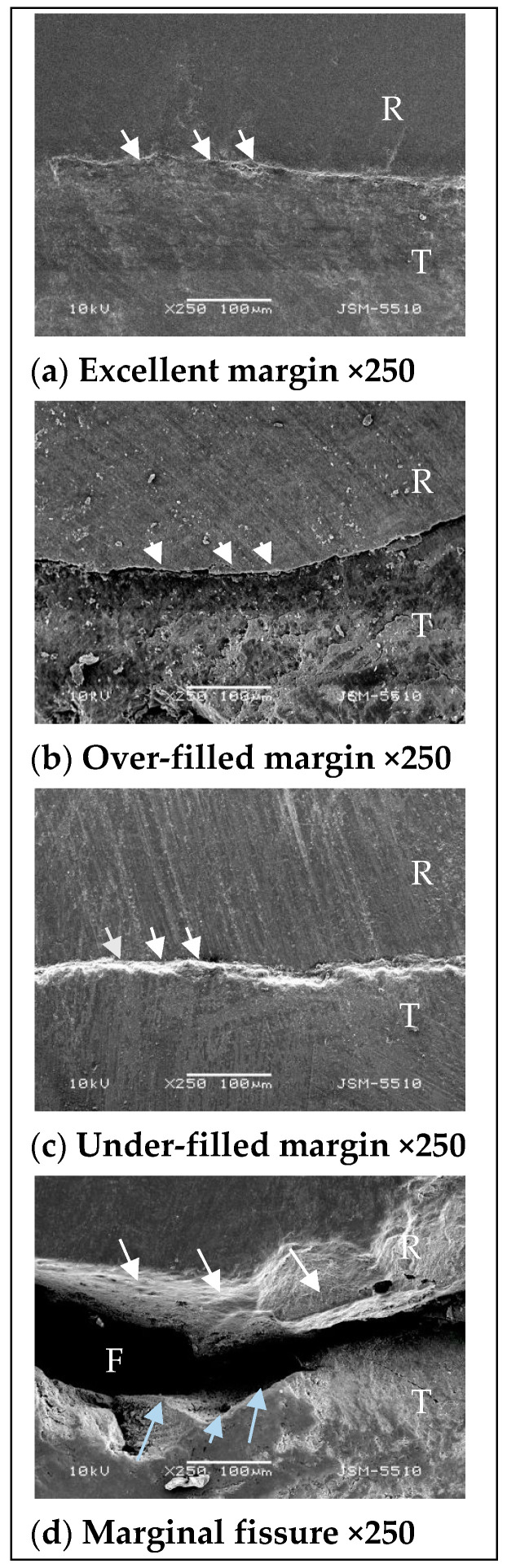
Criteria for assessing marginal adaptation, illustrated schematically by SEM images: R = restoration; T = tooth structure; F = fissure. (**a**) Excellent margin—arrows indicate the continuous restoration margin, covering precisely the preparation cavity margin; (**b**) over-filed margin—arrows indicate the continuous restoration margin hanging above the preparation cavity margin; (**c**) under-filled margin—arrows are pointing to the undercovered preparation cavity margin; (**d**) marginal fissure—white arrows are pointing to the restoration margin. Blue arrows are pointing to the cavity margin.

**Figure 3 polymers-13-01660-f003:**
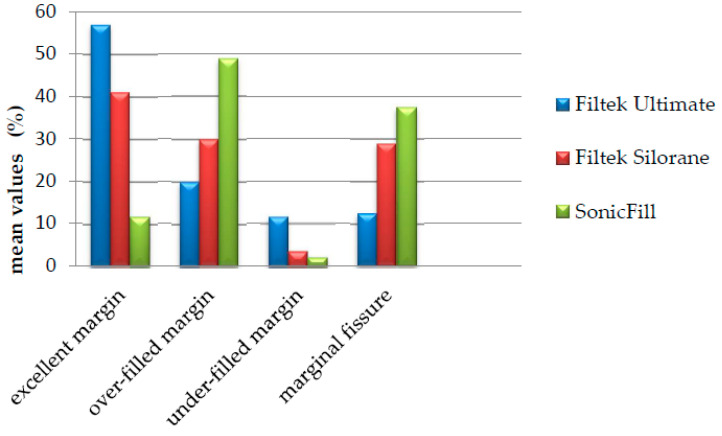
Marginal adaptation to the gingival margin -results, represented according to mean values.

**Figure 4 polymers-13-01660-f004:**
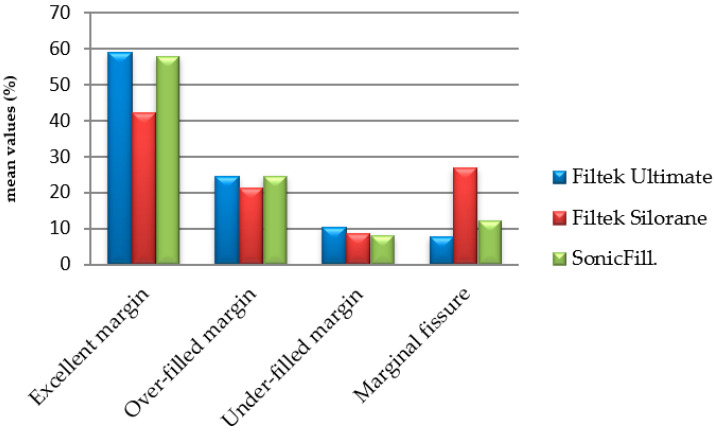
Marginal adaptation to the vestibular/lingual margins—results represented according to averaged mean values for both margins.

**Figure 5 polymers-13-01660-f005:**
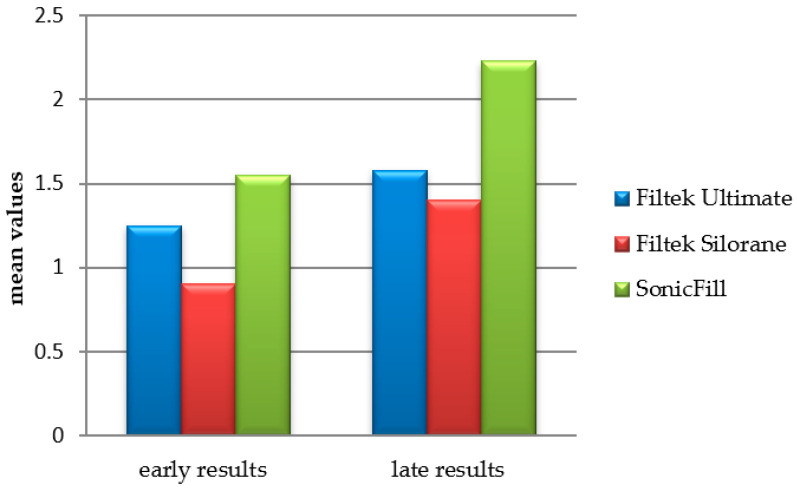
Descriptive comparison between early and late results of microleakage according to mean values.

**Figure 6 polymers-13-01660-f006:**
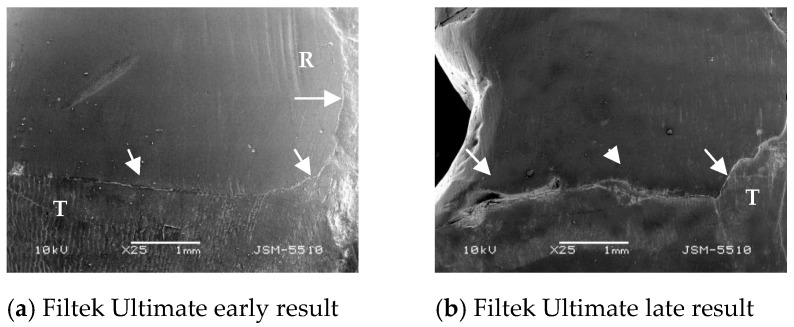
Illustrative comparison between early and late results of marginal adaptation (SEM); R = restoration; T = tooth structure; Arrows are pointing restoration margin. More discrepancies can be seen with the late results (**b**,**d**,**f**).

**Table 1 polymers-13-01660-t001:** Materials included in the study—composition.

Material	Organic Matrix	Material Type and Filler Loading
**Filtek Ultimate** (3M ESPE) A3 shade/body mass	Bis-GMA; TEGDMA; PEGDMA UDMA; Bis-EMA	Nanocomposite Filled 79 wt %
**Filtek Silorane LS** (3M ESPE) A3 shade	Silorane	Microhybrid Filled 76 wt %
**SonicFill** (Kerr) A3 shade	Bis-GMA; TEGDMA; UDMA; Bis-EMA	Nanohybrid, filled 84 wt % Bulk-fill

**Table 2 polymers-13-01660-t002:** Marginal adaptation to the gingival margin—results.

Values (%)			Excellent Margin	Over-Filled Margin	Under-Filled Margin	Marginal Fissure
	Materials	*n*	Mean ± sd	Mean ± sd	Mean ± sd	Mean ± sd
1	Filtek Ultimate	10	41.61 ± 27.04	20.50 ± 32.51	2.44 ± 7.72	35.37 ± 32.31
2	Filtek Silorane	10	41.24 ± 33.45	29.99 ± 26.88	3.54 ± 11.18	28.87 ± 18.88
3	SonicFill	10	11.70 ± 17.01	48.84 ± 31.50	2.08 ± 6.57	37.38 ± 26.43
**p. sign**			0.005	0.137	0.465	0.487

**Table 3 polymers-13-01660-t003:** Marginal adaptation to the proximal vestibular margin—results.

Values (%)			Excellent Margin	Over-Filled Margin	Under-Filled Margin	Marginal Fissure
	Materials	*n*	Mean ± sd	Mean ± sd	Mean ± sd	Mean ± sd
1	Filtek Ultimate	10	56.65 ± 29.88	19.52 ± 9.26	11.50 ± 21.81	12.43 ± 15.40
2	Filtek Silorane	10	41.01 ± 4.34	24.45 ± 6.37	6.94 ± 15.03	27.60 ± 19.52
3	SonicFill.	10	51.68 ± 6.27	23.01 ± 7.97	10.05 ± 12.02	21.47 ± 12.27
**p. sign**			0.012	0.397	0.243	0.156

**Table 4 polymers-13-01660-t004:** Marginal adaptation to the lingual proximal margin—results.

Values (%)			Excellent Margin	Over-Filled Margin	Under-Filled Margin	Marginal Fissure
	Materials	*n*	Mean ± sd	Mean ± sd	Mean ± sd	Mean ± sd
1	Filtek Ultimate	10	61.39 ± 19.36	29.55 ± 15.82	9.26 ± 16.65	3.07 ± 4.31
2	Filtek Silorane	10	44.37 ± 31.28	18.21 ± 18.04	10.25 ± 20.81	27.16 ± 39.63
3	SonicFill.	10	63.97 ± 11.57	26.41 ± 12.83	6.29 ± 15.42	3.33 ± 6.60
**p. sign**			0.025	0.376	0.298	0.024

**Table 5 polymers-13-01660-t005:** Results of micropermeability investigation.

Materials		Values											
		0		1		2		3		4		Total	
		*n*	%	*n*	%	*n*	%	*n*	%	*n*	%	*n*	%
1	Filtek Ultimate	7	17.5	15	37.5	10	25	4	10	4	10	40	100
2	Filtek Silorane	10	25	12	30	10	25	8	20	0	0	40	100
3	SonicFill	4	10	10	25	7	17.5	11	27.5	8	20	40	100
		***p*** ** = 0.005**											

**Table 6 polymers-13-01660-t006:** Gingival marginal adaptation. Comparison of early and late results.

Values (%)		Excellent Margin (Mean ± sd)			Over-Filled Margin (Mean ± sd)		Under-Filled Margin (Mean ± sd)		Marginal Fissure (Mean ± sd)	
	Materials	*n*	Early Results	Late Results	Early Results	Late Results	Early Results	Late Results	Early Results	Late Results
1	Filtek Ultimate	20	47.98 ± 25.95	41.61 ± 27.04	46.42 ± 23.30	20.50 ± 32.51	2.04 ± 6.55	5.41 ± 9.04	3.49 ± 6.10	35.37 ± 32.31
2	Filtek Silorane	20	55.78 ± 30.89	41.24 ± 33.45	36.51 ± 26.84	29.99 ± 26.88	1.67 ± 5.29	3.24 ± 10.23	6.02 ± 10.22	28.87 ± 18.88
3	SonicFill	20	27.85 ± 8.44	11.70 ± 17.01	49.90 ± 25.04	48.84 ± 31.50	5.38 ± 8.47	5.89 ± 11.94	21.84 ± 17.21	37.38 ± 26.43
	**p sign.**	***p* = 0.084**			***p* = 0.125**		***p* = 0.885**		***p* = 0.001**	

## Data Availability

Data sharing does not apply to this article.

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
