# Peer review of "Marginal Adaptation and Micropermeability of Class II Cavities Restored with Three Different Types of Resin Composites—A Comparative Ten-Month In Vitro Study"

_polymers, 2021, doi:10.3390/polym13101660_

Round 1

Reviewer 1 Report

Thank You very much for providing opportunity to review the manuscript titled “Marginal Adaptation and Micropermeability of Class II Cavities Restored with Three Different Types of Resin-Composites - A Comparative Ten-Month In Vitro Study

This work interesting and fit well with in the scope of this journal. In this study, authors have evaluated the marginal integrity and micropermeability in second-class cavities restored with three different types of composites. This is a well-designed study and the manuscript fits well within the scope of the journal; it needs some improvements; there are a few suggestions that authors may consider to improve it further:

The use of English language is reasonable, however, there are a number of punctuation and grammatical errors; that should be corrected and rephrased using academic English for a better flow of text for reader. The use of words such as “they, we, us” whould be reduced to minimum.

Abstract contains some unnecessary information such as background information may be reduces.

Introduction contains some unnecessary information. Authors missed some key references that would be important for a reader to access.

Line 89-90; please cite a reference for this information;

Polymerisation of restorative dental composites: influence on physical, mechanical and chemical properties at various setting depths." Materials Technology (2020): 1-7.

Similarly, line 96-98; please provide a citation if possible.

Ethics of this study is not clear from the “Materials and Methods” section, why intact molars were extracted? There might be some reasons for that. Did patients sign consent form before extractions?

Table 2: I surprise; why it is labeled as a table instead of figure. It should be figure 1;

Please add details of features in the captain; add a scale bar for SEM Images. Any point of interest may be indicated using arrow signs.

In the Discussion section, the results are not discussed from multiple angles. Author did not include limitations of the study and did not suggest any future research.

Some references are not cited according to journal guidelines, so they should be corrected.

In the conclusions, repetition should not be for “after 10 months of aging in saline” as it is also clear from the title.

Author Response

 The answers to reviewer 1 are attached as a PDF file.

Reviewer 2 Report

Dear authors,

You did a good work, but presented it poorly. First of all, this is a scientific manuscript, not tutorial! Numbering in the introduction is not acceptable. Moreover, short paragraphs can be combined to give the introduction a character of a scientific article. Based on subject, revisit paragraphs and make it modified. 

The following paper is missed in literature review:

https://www.sciencedirect.com/science/article/pii/S0300944019301262

The results should be revised. SEM micrographs are of low quality. They should be modified. Also some more images MUST be provided. The evolution during 10 months should be captured by SEM, hope that authors have this, otherwise it would be problematic to them! 

Authors must use graphs instead of several tables they used. The trends can be better seen in graphs. 

Discussion should be supported by an illustration in which mechanistic demonstration is used. Authors should visualize from molecular view what happened during 10 months. The early and late stages must be compared mechanistically in that illustration. 

Conclusion is too short! It should be both quantitative and qualitative. No quantities is used, no comparison is made based on results. Discussion should also be added to that part. 

I suggest major revision. 

Author Response

The answers to reviewer 2 are attached as a PDF file.

Round 2

Reviewer 1 Report

Many thanks for the revision and incorporating all suggested changes to the manuscript

Author Response

To Reviewer 1

Many thanks for the revision and incorporating all suggested changes to the manuscript

Response:

Thanks for the positive review. Thank you for appreciating our efforts to improve the quality of the manuscript exposition.

(x) English language and style are fine/minor spell check required

We appreciate your opinion on our efforts to improve our English.

*The response is attached like a PDF file too.

Reviewer 2 Report

Authors responded to my comments selectively. More SEM should be added. discussion should be supported by schematic mechanism. A reference missed should be added ....

This is not the way authors should take in responding to answers, I again suggest major revision. Authors must see my first report and revise the manuscript. 

Author Response

Response to Reviewer 2 Comments

Authors responded to my comments selectively. More SEM should be added. discussion should be supported by schematic mechanism. A reference missed should be added ....

This is not the way authors should take in responding to answers, I again suggest major revision. Authors must see my first report and revise the manuscript. 

We are very sorry that we did not structure our answers according to the template provided by the magazine. Therefore, for the first time, we have included responses to all the comments you have included. We will now answer again by adding our reactions to the new observations.

Point 1: You did a good work, but presented is poorly. First of all, this is a scientific manuscript, not tutorial! Numbering in the introduction is not acceptable. Moreover, short paragraphs can be combined to give the introduction a character of a scientific article. Based on subject, revisit paragraphs and make it modified.

The following paper is missed in literature review: https://www.sciencedirect.com/science/article/pii/S0300944019301262

Response 1: 1. The introduction has been changed to avoid "the tutorial style". The paragraphs have been expanded and supplemented. We have not supplemented the references with the one suggested by you (https://www.sciencedirect.com/science/article/pii/S0300944019301262)  as it is not related to the presented topic.

Point 2: The results should be revised. SEM micrographs are of low quality. They should be modified. Also some more images MUST be provided. The evolution during 10 months should be captured by SEM, hope that authors have this, otherwise it would be problematic to them!

Response 2: Results. They are expanded with comparative information based on our previous study, conducted under the same conditions and results reported immediately after thermal cycling. Thanks for a good idea.

- SEM are not of poor quality. The original format is 1280 x 960 pixels. Here they are compressed in web format.

- Figures have been added to the presentation of the results in order to better illustrate the information. The tables are preserved because they show more correctly the data included in the statistical analyzes.

- A figure representing the sequence of the experiment is included (Fig. 1)

Point 3. Discussion should be supported by an illustration in which mechanistic demonstration is used. Authors should visualize from a molecular view what happened during 10 months. The early and late stages must be compared mechanistically in that illustration.

Response 3: The discussion has been supplemented and expanded. Articles have been used to clarify our conclusions and results in light of the findings and publications of other researchers.

 The analyzes of marginal adaptation and micropermeability are a quantitative interpretation of qualitative indicators. The reported results are presented in detail in their quantitative dimensions in the results section. In the discussion, we have included reflections on the achieved results, which we find essential.

Point 4.Conclusion is too short! It should be both quantitative and qualitative. No quantities is used, no comparison is made based on results. Discussion should also be added to that part.

 Response 3: We believe that the conclusions reflect the important results of the study. They are supplemented by another one, based on the comparative analysis of the early and late outcomes.

New

Poin 1:  More SEM should be added. discussion should be supported by schematic mechanism.

Response 1: We've reviewed your previous recommendations. For the first time and now we have tried to improve the article presentation, considering your suggestions. Now we have included Figure 6, which is an illustration of early and late results. It complements and supports the discussion. Also, it includes SEM images,  illustrating the changes that occur during ageing.

The discussion has its scheme. First: commented on the effect of ageing on recovery. Comments focus on the gingival margin as a weak point of second-class restorations. The marginal adaptation/microleakage relationship is commented on. All judgments are supported by evidence from the study. All results were compared and reviewed in comparison with other published studies.

Second: The materials themselves are considered, one by one. Their results are explained by specific characteristics that can impact the results registered by us (shrinkage, modulus of elasticity, sorption, solubility, etc.). Finally, comparisons of early and late results are made.

Third: In the conclusion part, the efforts and difficulties in developing an ideal and durable material are commented on. The limitations resulting from the current in vitro study are indicated. Ideas for new and additional research are given.

 Fourth: Conclusions: We believe that they cannot be too generalizing because each commercial product has its characteristics that affect its qualities. It bears the features of the group, but also the specific development like a commercial product. Therefore, the conclusions reflect the results very specifically.

   We believe that the article follows the structure of scientific research published in the world literature on marginal adaptation and micropermeability. The analysis of marginal adaptation and micropermeability is a quantitative interpretation of qualitative indicators trying to recreate and evaluate clinical features. The reported results are presented in detail, and the critical comparisons between early and late results are made, leading to the main conclusions of the work.

Point 2: A reference missed should be added ....

We cannot add the missing reference. https://www.sciencedirect.com/science/article/pii/S0300944019301262:

Electroactive bio-epoxy incorporated chitosan-oligoaniline as an advanced hydrogel coating for neural interfaces.SaeedManouchehriaBabakBagheribSomayeh HosseiniRadcMojtaba NasiriNezhaddYeu ChunKimbO. OkParkbMehdiFarokhieMaryamJouyandehfgMohammad RezaGanjalifiMohsen KhodadadiYazdiaPayamZarrintajcfhMohammad RezaSaebfhj

Abstract: Neural electrodes have opened new horizon in neuroscience which discover the new aspect of neural and brain behavior. Neural electrodes are usually fabricated using metals. Metals cause chronic inflammation in long term due to mechanical mismatch with soft tissue. In this regards, hydrogel based coating has been proposed to regulate the modulus. Electroactive hydrogel coating because of the similar modulus with brain, proper biocompatibility and low impedance can be used as proper coating for neural interface. In this work, the electroactive epoxidized chitosan was firstly synthesized and studied for electroactivity using cyclic voltammetry and UV–vis. Cyclic voltammetry diagram exhibited two redox peak between 0.38 and 0.8 related to oxidation/reduction transient; In addition, UV–vis spectra exhibited π-π* transition and benzenoid to quinoid excitonic transition around 330 nm and 620 nm, respectively. Swelling ratio of the electroactive epoxidized chitosan was decreased 150% compared to electroactive one (500%). The hydrophobic oligoaniline decreased the swelling ratio and hindered the water molecule diffusion within the hydrogel structure. Hence, the degradation rate was decreased by enhancing the oligoaniline content. Electroactive coating showed the lower impedance compared to isolated one. Moreover, the electroactive substrate exhibited more cell proliferation because of its conductive nature which enhanced the PC12 cells activity. This study introduced the novel hydrogel based on electroactive coating, which can be potentially used as a neural electrode coating and pave a way for architecting new coating for biomedical devices.

Keywords: Oligoaniline; Bio-epoxy; Chitosan; Neural electrode; Electroactive; Coating introduction

 It is not related to the current topic.

Point 3:(x) Extensive editing of English language and style required

In your first review, you stated (x)  Moderate English changes required.

We did a lot for English. We worked together with a language specialist.

We believe that the style has improved. We do not think that language corrections have led to a deterioration in English writing.

* Response is attached as a PDF file too.
